# Characterization and prediction of individual functional outcome trajectories in schizophrenia spectrum disorders (PREDICTS study): Study protocol

Sri Mahavir Agarwal[1,2,3,4], Joel Dissanayake[1], Ofer Agid[1], Christopher Bowie[1], Noah Brierley[1], Araba Chintoh[1], Vincenzo De Luca[1], Andreea Diaconescu[1], Philip Gerretsen[1], Ariel Graff-Guerrero[1], Colin Hawco[1], Yarissa Herman[1], Sean Hill[1], Kathryn Hum[1], Muhammad Omair Husain[1], James L. Kennedy[1], Michael Kiang[1], Sean Kidd[1], Nicole Kozloff[1], Marta Maslej[1], Daniel J. Mueller[1], Farooq Naeem[1], Nicholas Neufeld[1], Gary Remington[1], Martin Rotenberg[1], Peter Selby[1], Ishraq Siddiqui[1], Kate Szacun-Shimizu[1], Arun K. Tiwari[1], Shanthos Thirunavukkarasu[1], Wei Wang[1], Joanna Yu[1], Clement C. Zai[1], Robert Zipursky[1], Margaret Hahn[1,2,3,4‡]*, George Foussias[1,2,3‡]*

1 Schizophrenia Division, Centre for Addiction and Mental Health (CAMH), Toronto, Canada, 2 Temerty Faculty Institute of Medical Science, University of Toronto, Toronto, Canada, 3 Department of Psychiatry, University of Toronto, Toronto, Canada, 4 Banting and Best Diabetes Centre (BBDC), University of Toronto, Toronto, Canada

‡ MH and GF are shared senior authors on this work.
* margaret.hahn@camh.ca (MH); george.foussias@camh.ca (GF)

**Data Availability Statement:** N/A- protocol reports no results.

## Abstract

Schizophrenia spectrum disorders (SSDs) are associated with significant functional impairments, disability, and low rates of personal recovery, along with tremendous economic costs linked primarily to lost productivity and premature mortality. Efforts to delineate the contributors to disability in SSDs have highlighted prominent roles for a diverse range of symptoms, physical health conditions, substance use disorders, neurobiological changes, and social factors. These findings have provided valuable advances in knowledge and helped define broad patterns of illness and outcomes across SSDs. Unsurprisingly, there have also been conflicting findings for many of these determinants that reflect the heterogeneous population of individuals with SSDs and the challenges of conceptualizing and treating SSDs as a unitary categorical construct. Presently it is not possible to identify the functional course on an individual level that would enable a personalized approach to treatment to alter the individual's functional trajectory and mitigate the ensuing disability they would otherwise experience. To address this ongoing challenge, this study aims to conduct a longitudinal multimodal investigation of a large cohort of individuals with SSDs in order to establish discrete trajectories of personal recovery, disability, and community functioning, as well as the antecedents and predictors of these trajectories. This investigation will also provide the foundation for the co-design and testing of personalized interventions that alter these functional trajectories and improve outcomes for people with SSDs.

**Funding:** This study is funded by the CAMH Discovery Fund. The funders did not and will not have a role in study design, data collection and analysis, decision to publish, or preparation of the manuscript.

# 1. Introduction

## 1.1. Functional disability and recovery in schizophrenia spectrum disorders

Schizophrenia spectrum disorders (SSDs) including schizophrenia, schizoaffective disorder, and other primary psychotic disorders are serious and persistent mental illnesses characterized by a combination of positive symptoms (i.e., delusions and hallucinations), cognitive impairment (including deficits in attention, working memory, and executive function), and negative symptoms (i.e., diminished emotional expression and motivation deficits [1]. Estimates suggest that the lifetime prevalence of SSDs in the population is up to 2% [2]. Despite some advances in treatment of SSDs, enduring impairments in community functioning continue to be a hallmark of the disorder with a median recovery rate of 13.5% [3–5]. Therefore, SSDs remain a leading cause of disability worldwide [6, 7] and are associated with tremendous economic and personal costs. These include lower rates of employment [8, 9], educational achievement [10], and social and romantic relationships [11], contributing to lost productivity [8, 12, 13]. This burden also enormously impacts the wellbeing of family members and caregivers [14].

In addition to disability quantified by objective measures, individuals with SSDs also experience subjective impairments in quality of life, physical health, mood, leisure activities, and social relationships [15, 16], although substantial inter-individual variability exists [17–20]. However, previous longitudinal studies of functional outcomes in SSD have not focused on the critically important domain of subjective recovery [21–24]. Person-centered or personal recovery can be conceptualized as a set of processes and/or outcomes that support a personally meaningful life in the context of having a mental illness [25]. This incorporates key personal factors including connectedness, hope, identity, meaning, and empowerment [26]. Although stakeholders have increasingly embraced this conceptualization of personal recovery [27], there has been limited research on longitudinal trajectories of recovery [28] and there are gaps in the literature regarding concepts such as community function and disability related to personal recovery over time [29].

## 1.2. Predictors of functional disability and recovery

**1.2.1. Psychopathology.** Psychotic symptoms are tremendously disabling [6, 7] and impair psychosocial functioning considerably. As predictors of functional disability, both positive and negative symptoms have been associated with lower likelihood of meeting criteria for clinical recovery in later phases of illness [30–33]. Negative symptoms have generally demonstrated greater association with psychosocial functioning in SSD compared to positive symptoms [34–41] and predict functional impairment through the course of the illness [39–48], regardless of whether negative symptoms are broadly defined or explicitly restricted to primary negative symptoms [49, 50].

Cognitive deficits have also been identified as key determinants of functional outcomes [51, 52], although the size of this effect remains uncertain [35, 42, 53, 54]. With regard to domains of functioning, social cognition may be most closely associated with social functioning [55]. The theory of mind, social perception and knowledge have emerged as specific cognitive domains that are strongly associated with psychosocial functioning [56, 57]. Meta-analytic findings suggest that social cognition accounts for more unique variance in functioning than neurocognition [56, 57]. In addition, the relationship between neurocognition and functioning appears to be at least partially mediated by social cognition [58–64].

Beyond the traditional symptoms, depression, anxiety, and substance abuse have been associated with longitudinal subjective quality of life along with other psychosocial measures

including self-efficacy and social support perceived from significant others [20, 65]. While understudied compared to objective measures of community functioning, recent studies have indicated that negative emotions including depression, anxiety, negative self-esteem, and hopelessness, as well as internal locus of control have emerged as central predictors of personal recovery [66–68]. Similarly, substance use has been identified as a predictor of personal recovery. Recent findings indicate that the misuse of substances, specifically alcohol and marijuana, are often associated with poor outcomes regarding independent living and social compliance [69, 70]. In addition, this disorder has been shown to be a primary cause of frequent emergency department visits, red-admission into inpatient care, and incarceration [69, 70].

**1.2.2. Genetic and biological predictors.** Twin, family, and adoption studies support a strong genetic component in the risk of developing SSDs, with heritability estimates in the range of 80–87% [71, 72]. Among the SSDs, schizophrenia (SCZ) is a severe and most commonly studied disorder. A Hypothesis-free genome wide association study (GWAS) of SCZ using over 76,755 patients and over 243,649 controls identified 342 independent single nucleotide polymorphisms (SNPs) at 287 distinct genomic loci to be significantly associated with SCZ. However, the clinical utility of each individual common variant in diagnosing SCZ is very small. To this end, genome-wide polygenic risk scores (PRS), generated from thousands of GWAS risk variants, has emerged as an important tool that may in the future identify individuals at significant risk for SCZ before the illness manifests. PRS analysis has emerged as an important tool to potentially predict outcomes in mental health [73]. Recent work has correlated PRS with Global Assessment of Functioning (GAF) scale scores pre-treatment [74], and PRS explain a significant, albeit small, proportion of variance in quality of life in people with SSDs above and beyond demographic and clinical variables [75]. There is also an emerging literature on brain imaging predictors of functional outcomes in patients with SSDs. Historically enlarged lateral ventricles have been associated with poor functional outcomes [76], However, more regional findings through longitudinal magnetic resonance imaging (MRI) have attributed poor functional outcomes to volumetric decreases in frontal lobe structures [77], specifically the inferior, middle, and superior frontal gyri [78].

**1.2.3. Physical health.** Physical health and metabolic comorbidities have also been identified as predictors of functional disability amongst individuals living with SSD. Individuals with SSD have significantly higher rates of cardiometabolic diseases, including type 2 diabetes (T2D), respiratory diseases, liver diseases, cancers, and lower physical fitness [79, 80], leading to early mortality [79, 81, 82]. Despite this burden, access to metabolic monitoring, physical health care, and interventions remain suboptimal relative to the general population [83, 84]. Concerningly, findings suggest that the mortality gap from CVD in SSDs may be increasing over time relative to the general population [85, 86]. Beyond cardiovascular health, metabolic comorbidity is also associated with poorer quality of life [87], stigma [88], barriers to social engagement [89] and poorer adherence with treatment [90], all contributing to poorer mental health outcomes. Furthermore, physical health and other metabolic factors could represent modifiable risk factors for domains such as cognition that are classically recalcitrant to treatment [91–93].

**1.2.4. Social predictors.** Socio-environmental exposures at both the individual and environmental level have been established as risk factors in developing a SSD. In addition, the extension of these known risk factors on functioning, disability and recovery is an area of increasing interest. Some of the most compelling evidence for socio-environmental risk comes from well-replicated findings of increased incidence of SSD in immigrant and minority ethnic communities [94–97], in urban settings [98], and in the context of childhood trauma, discrimination [99], social isolation and loneliness [100], as well as employment and achievement-expectation mismatch [101]. In particular, racialized people with a SSD have been found to

experience worse clinical, social and service use outcomes in previous cohort studies [96]. In contrast to these risk factors, psychological resilience [102] and social capital [103] have been identified as potential protective factors that may buffer the impact of detrimental socio-environmental exposures and support positive outcomes. Beyond the ongoing impact these risk factors have on the course of SSD and outcomes, people with a SSD must also contend with the impacts of stigma, social exclusion, challenges in accessing services and care, which may further disparities [96].

## 1.3. Outcomes for individuals with SSDs are heterogeneous, with distinct trajectories of functioning within population subsets

An examination of the heterogeneity of symptoms and functioning for individuals with SSD has historically identified up to 8 different course trajectories varying in type of onset, course type, and end state [104, 105]. More recently, studies have used data-driven modeling approaches to examine longitudinal trajectories of functional outcomes in SSDs within the first few years of illness, finding that those in higher social functioning trajectories were more likely to be female, have higher cognitive functioning, less substance abuse history, and better premorbid functioning in late adolescence [21, 106]. Long-term data-driven longitudinal studies have reported four outcome trajectories of social functioning and improvement in individuals with SSD: 1) preserved, 2) moderately impaired, 3) severely impaired, and 4) profoundly impaired functioning [24, 107]. In addition, initial comparisons of longitudinal functioning amongst individuals with SSDs to those with mood disorders indicated a substantial and consistent decline in functioning over time for people with SSDs [108].

Overall, while these studies provide a more nuanced understanding of the heterogeneous longitudinal functional trajectories for individuals with SSDs, several gaps remain. In many instances, outcome trajectories were based on biomedical frameworks rather than personal recovery or self-reported disability. In addition, most studies used a limited set of sociodemographic and clinical variables, physical health examinations, and neurobiological markers. Consequently, this may impact the ability to adequately establish long-term trajectories and implement treatment interventions at the individual patient level [26, 109].

## 1.4. Summary and rationale

Despite many decades of research and efforts at treatment innovation, people with SSDs continue to experience significant impairments across important outcome domains, including personal recovery, physical and mental health related disability, and community functioning. To date, most attempts to characterize functional trajectories in SSDs have utilized population-based approaches to identify a broad range of demographic, social, environmental, clinical, physical health, and biological predictors of functional outcomes. However, findings across studies suggest that such effects are not in fact uniform, but rather depend on the population sampled and the underlying trajectories comprising the sampled population. Similar challenges have emerged with regards to findings for treatment interventions. Most studies have either focused on treating a specific domain of psychopathology to mediate improvements in functioning, or broader health and psychosocial interventions that are delivered to individuals based on their diagnosis of an SSD. Within a biomedical framework, recovery and functional outcomes in SSDs typically focus on symptom management and indices of social and role functioning [5, 110]. There is a scarcity of research evaluating the longitudinal trajectories of personal recovery and disability in people with SSDs, and how these outcomes interact with each other and with traditional measures of community functioning over time.

In order to advance the understanding of outcome trajectories and potential treatment intervention opportunities, there is a need for a broader and more comprehensive evaluation of discrete trajectories of personal recovery, disability, and community functioning in individuals with SSDs. To our knowledge, no longitudinal study has examined these concurrent trajectories, nor incorporated the impact of physical health, neurobiology, and genetic predictors on these trajectories. The proposed study will advance understanding of unique and shared longitudinal courses, as well as underlying clinical, socio-environmental, physical health and biological predictors. In addition, the inclusion of electronic medical record (EMR)-based analytics and administrative health data provides additional unique opportunities for expanded clinical phenotyping and longitudinal outcome evaluations. The proposed study will also set the stage for the co-design and testing of real-world interventions targeting discrete outcome trajectory subtypes through a precision medicine approach. Findings from this research may help improve outcome trajectories, promote recovery, and reduce the enduring disability of individuals with SSD.

## 2. Objectives and hypotheses

**Objective 1:** Determine the longitudinal functional trajectories of individuals with SSDs across three co-primary domains consisting of personal recovery, disability, and community functioning.

**Hypothesis 1a:** Within each outcome domain, we anticipate approximately 4–5 longitudinal trajectories, consisting of early-sustained improvement, delayed gradual improvement, early improvement / delayed mild-moderate deterioration, early improvement / delayed moderate-severe deterioration, and persistent disability.

**Hypothesis 1b:** We hypothesize that early improvement in personal recovery trajectories will be antecedents of subsequent improvement in disability and community functioning trajectories, while deteriorating or persistent disability, particularly with regards to physical disability, will be early antecedents of worse community functioning.

**Objective 2:** Develop and test predictive models to accurately determine functional trajectories at the individual level.Hypothesis 2a

Trajectories of functioning will be differentiable based on sociodemographic and treatment factors, psychopathology, physical health, and biological measures.

**Hypothesis 2b:** A subset of these differential baseline sociodemographic, symptoms, and biological measures will enable the prediction of individual functional trajectories with high positive predictive power.

**Exploratory Objective:** Develop and validate a novel automated method using natural language processing (NLP) and machine learning (ML) for interrogating unstructured EMRs, to ascertain retrospective temporal trajectories in symptom, physical health, treatment, and functional outcomes.

**Exploratory Hypotheses:** NLP-ML-based characterization of temporal trends in functional outcomes, symptoms, physical health, and treatment will be correlated with relevant structured clinical measures and differentiate the longitudinal outcome trajectory subtypes identified in the first objective. Thus, NLP-ML-based temporal trends in functioning will serve as antecedent markers to inform the discovery of prospective outcome trajectories.

## 3. Methodology

This study will utilize a longitudinal sequential cohort design, with broad sampling across age ranges and phases of illness for individuals with SSDs. The characterization and prediction of trajectories of functional disability will provide the foundation for future integration of

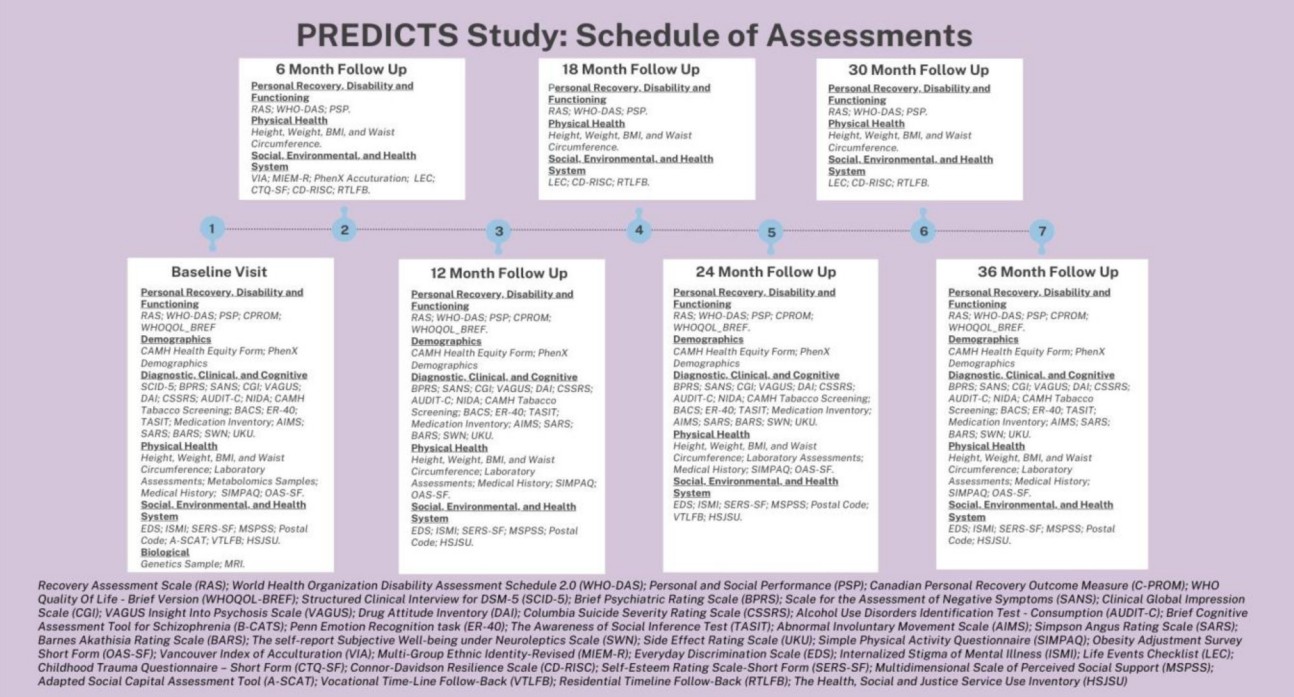

**Fig 1. Schedule of study assessments.**

interventions targeting specific predictors of functional disability that will be co-designed with patients, family members, and service providers, and evaluated as embedded trials within this cohort study. Outcomes, measures, and assessments were selected to comprehensively capture relevant domains of functional outcomes and recovery. This selection was informed by patient and family member representatives of the study's steering committee, as well as their integration within routine clinical services at CAMH and alignment with other local, national, and international clinical research initiatives. Study assessments will be completed at baseline and longitudinally, as outlined in **Fig 1**. Extended follow-up beyond the initial three years will be pursued through external funding to support this work. Passive follow-up extending beyond the clinical follow-up period will also be possible through linkage with health administrative data.

### 3.1. Participants

The study will recruit 1000 participants across all outpatient and inpatient clinical services at CAMH providing care for individuals with SSDs and related disorders. The Rationale for choosing the given sample size is to provide more precise effect size estimates that can be potentially utilized for clinically useful risk stratification. In addition, previous studies have attempted to examine outcomes and functional trajectories in diverse samples of people with SSD but have been limited by study quality and sample size [111].

**3.1.1. Inclusion criteria.** Participants will be eligible for this study if they: 1) are 16 years of age and older; 2) have a DSM-5 diagnosis of any schizophrenia spectrum disorder (SSD) or other disorders with psychotic features such as bipolar I disorder with psychotic features, major depressive disorder with psychotic features, and substance/medication-induced psychotic disorder; and 3) have adequate fluency in English to participate in clinical care without

the need for a translator. Of note, participants will be eligible regardless of duration of illness, co-morbid mental or physical health conditions, capacity to consent to treatment, or whether they are receiving inpatient or outpatient services.

**3.1.2. Exclusion criteria.** For this study, there are no specific exclusion criteria. Therefore, any individual that meets the inclusion criteria will be eligible to participate.

These broad eligibility criteria will serve to support generalizability of the sample. In addition, the inclusion of participants with other related psychiatric disorders aside from those with SSD's will maximize the opportunity to evaluate the influence of common co-occurring illnesses, and enable an accelerated longitudinal design maximizing coverage of longitudinal functional trajectories across illness stages.

**3.1.3. Consent process.** Participants will provide written informed consent. During the consent process, we will use a structured framework to assess the capacity of participants to consent to the study or whether a substitute decision maker (SDM) is needed. For individuals deemed incapable to consent to treatment, their SDM will provide written informed consent in conjunction with participant assent to study participation.

## 3.2. Procedures

Participants will undergo comprehensive clinical and functional characterization using a series of standard measures, including structured diagnostic assessment, clinical assessments to index severity of psychiatric symptoms, substance use, physical health, cognition and functioning, along with collection of treatment and medical history data from participants and their EMR (Table 1). Demographic information will be collected through routine use of the CAMH Health Equity Form [112] in addition to revised items from the PhenX Toolkit Demographic Protocol [113] and further characterization of social and health equity factors. Linkage is planned with broader provincial health administrative data held at ICES (formerly known as the Institute for Clinical Evaluative Sciences. Participant assessments will be conducted either in-person or virtually, where possible, based on participant preference. All participants will be offered the opportunity to participate in the neuroimaging and laboratory/biological components of this study.

## 3.3. Outcome measures

**3.3.1. Primary measures.** To evaluate the outcomes of disability, recovery, and community functioning, three primary measures will be employed: 1) The 22 item version of the *Recovery Assessment Scale (RAS)* [114]; 2) *World Health Organization Disability Assessment Schedule 2.0 (WHO-DAS)* [115]; and 3) *Personal and Social Performance scale (PSP)* [116, 117].

Literature identifies the 22-item version of the RAS as one of the most widely used measures of person-centered recovery. With well-established psychometric properties, this measure has indicated good validity and reliability in 49 and 19 studies, respectively [114]. The WHO-DAS is a 36-item measure of disability and health that is grounded in the framework of the WHO's International Classification of Functioning, Disability and Health [115]. Amongst different populations, this measure enables a comprehensive evaluation of individuals' functioning across six major life domains (cognition, mobility, self-care, getting along, life activities, and participation in society). In addition, literature indicates this measure to have good psychometric qualities through good reliability, concurrent validity, a stable factor structure across populations, and is sensitive to change over time [115]. For this study, we will employ both the participant self-report version as a primary outcome measure, and, where available, will also seek input from the informant-version of the WHO-DAS. To capture long-term trajectories of community functioning, we will be utilizing the PSP, an anchored version of the Social and Occupational Functioning

**Table 1. List of clinical assessments for secondary outcome measures.**

| Outcome Measures | Clinical Assessments and Internal Reliability (Cronbach's alpha) |
|---|---|
| Recovery and Quality of Life | 1) Canadian Personal Recovery Outcome Measure (C-PROM) [118] **(Good Internal Validity)** [118]<br>2) WHO Quality of Life Brief Version (WHOQOL-BREF) [119] **(0.896)** [120] |
| Psychopathology (Clinical Measures) | 1) Structured Clinical Interview for DSM-5 (SCID-5) [121] **(N/A)**<br>2) The 24-item Brief Psychiatric Rating Scale (BPRS) [122] **(0.87)** [123]<br>3)Scale for the Assessment of Negative Symptoms (SANS) (Andreasen, 1982) [124] **(0.885)** [125]<br>4) Clinical Global Impression (CGI) scale [126] **(N/A)**<br>5) VAGUS insight into psychosis scale [127] **(0.745)** [127]<br>6) Drug Attitude Inventory (DAI) [128] **(0.889)** [129]<br>7) Columbia Suicide Severity Rating Scale (CSSRS) [130] **(0.937)** [131]<br>8) Modified Colorado Symptom Index (mCSI) [132] **(0.90)** [132] |
| Substance Use | 1) Alcohol Use Disorders Identification Test—Consumption (AUDIT-C) [133] **(0.96)** [134]<br>2), NIDA Quick Screen/Modified ASSIST tool [135] **(N/A)**<br>3) CAMH Tobacco Screening Tool **(N/A)** |
| Cognition | 1) Brief Assessment of Cognition in Schizophrenia (BACS) [136] **(Good Internal Reliability)** [136]<br>2) Penn Emotion Recognition task (ER-40) (Kohler et al., 2000) [137] **(0.808)** [138]<br>3) The Awareness of Social Inference Test—Short (TASIT-S) [139] **(Good Internal Reliability)** [139]. |
| Treatment History and Medication Side Effects | 1) PhenX Toolkit medication inventory **(N/A)**<br>2) Abnormal Involuntary Movement Scale (AIMS) [140] **(Good Internal Reliability)** [141]<br>3) Simpson Angus Rating Scale (SARS) [142] **(0.79)** [143]<br>4) Barnes Akathisia Rating Scale (BARS) [144] **(Good Internal Reliability)** [144]<br>5) Subjective Well-being under Neuroleptics (SWN) scale [145] **(0.93)** [146]<br>6) UKU Side Effect Rating Scale–Self Report [147] **(Good Internal Reliability)** [147] |
| Physical Health | 1) Personal/Family History of cardiovascular disease (CVD), Lifestyle review (diet, activity, smoking) **(N/A)**<br>2) Weight and waist circumference. BMI is calculated (weight/height $^2$) **(N/A)**<br>3) Heart rate and blood pressure **(N/A)**<br>4) Simple Physical Activity Questionnaire (SIMPAQ) [148] **(Good Internal Reliability)** [148]<br>5) Obesity Adjustment Survey-Short Form (OAS-SF) [149] **(0.719)** [149]<br>6) WHO Quality of Life Brief Version (WHOQOL-BREF) [119] **(0.896)** [120] |
| Social, Environmental, and Health System Measures | 1)Vancouver Index of Acculturation–Short (VIA) [150] **(0.87)** [150]<br>2) Multi-Group Ethnic Identity-Revised (MIEM-R) [151] **(0.88)** [151]<br>3) 8 item PhenX Acculturation Survey [113] **(N/A)**<br>4) Everyday Discrimination Scale (EDS) [152] **(0.80)** [153]<br>5) The Internalized Stigma of Mental Illness (ISMI) [154] **(0.94)** [155]<br>6) Life Events Checklist (LEC) [156] **(Good Internal Validity)** [156]<br>7) Childhood Trauma Questionnaire–Short Form (CTQ-SF) [157] (0.97) **[157]**<br>8) Connor-Davidson Resilience Scale [158] **(0.89)** **[158]**<br>9) Self-Esteem Rating Scale-Short Form (SERS-SF) [159] **(0.90)** [159]<br>10) Multidimensional Scale of Perceived Social Support (MSPSS) [160] **(0.95)** [161]<br>11) Adapted Social Capital Assessment Tool (A-SCAT) [162] **(0.95)** [163]<br>12) Vocational Time-Line Follow-Back (VTLFB) [164] **(N/A)**<br>13) Residential Timeline Follow-Back (RTLFB) [165] **(0.88 for 6 months)** [166]<br>14) Health, Social and Justice Service Use Inventory (HSJSU) [167] **(N/A)** |

Assessment Scale (SOFAS) from the DSM-IV. This measure allows for separate evaluations across community functioning domains for individuals with SSDs. Such domains include: role functioning, social functioning, self-care, and disturbing and aggressive behavior [116]. In addition to its diverse evaluation of community functioning, the PSP has indicated good test-retest and inter-rater reliability, validity, and sensitivity to change over time [116, 117].

**3.3.2. Secondary measures.** *3.3.2.1. Clinical assessments.* In addition to the primary measures, a broad assessment approach will be utilized to evaluate the complementary constructs of personal recovery, disability and community functioning. This study will utilize a combination of both participant self-report and rater-administered clinical assessments. These assessments will be employed to evaluate the following outcome measures: recovery and quality of life; psychopathology; substance use; cognition; treatment history and medication side effects; physical health; and social, environmental, and health system measures. A full list of the clinical assessments with reliability scores that will be used is listed in **Table 1**.

*3.3.2.2. Blood-based measures.* Fasting blood work will be collected at baseline to measure glucose, insulin, HbA1c, CRP, lipid profile, and liver, kidney, and thyroid function. HOMA-IR, an index of insulin resistance will be calculated from fasting glucose and insulin levels. These assessments will be repeated yearly unless abnormal. Fasting glucose, insulin, HbA1c, and lipids will be repeated at 3 months, if values are abnormal, if starting or switching a new antipsychotic medication, and/or initiating adjunctive pharmacological intervention to target metabolic comorbidity. In addition to the above laboratory assessments related to physical health, study participants will be offered the opportunity to provide samples for future genomic and metabolomic analyses.

For participants interested in participating in our genomic research arm, genomic DNA will be purified from blood or saliva samples and genotyped on Global Diversity Arrays (Illumina). Calculations of the PRS score will follow the current best practices [168] using the GWAS summary statistics (e.g. [169] for SCZ from patients of European, mixed-ancestry meta-analyses and other ancestries. A recent comparison of PRS methods suggests that methods such as MegaPRS, LDPred2 and SBayesR gave the highest prediction statistics compared to the conventional p-value based clumping and thresholding method (e.g. PRSice) in individuals with psychiatric disorders [170]. We will explore the use of methods such as PRSice2 [171], PRS-CS [172], LDpred2 [173], SBayesR [174], and multi-PRS [175].

*3.3.2.3. Neuroimaging measures.* Magnetic resonance imaging (MRI) will be conducted at baseline on a 3 Tesla GE Discovery 750 scanner. Standardized multi-band imaging sequences as used in the Adolescent Brain Cognitive Development (ABCD) cohort study will be used to acquire T1-weighted, multi-shell diffusion MRI, and resting state fMRI (R-fMRI) data; key acquisition parameters have been published previously [176]. Given the importance of visceral fat and its impact on cardiovascular health, an abdominal T1-weighted image and an image to allow for water-triglyceride fat separation (IDEAL-IQ) will be acquired using the following parameters: repetition time/time to echo (TR/TE) = 5.768 ms/2.64 ms, slice thickness = 10 mm, Spacing between slices = 10, Echo train length = 3, Acquisition matrix = 160, Reconstruction matrix = 256, Pixel bandwidth = 868.047, field of view (FOV) = 44, flip angle = 3˚, number of excitation (NEX) = 0.5 [177–180].

*3.3.2.4. Health administrative data.* All participants in the study will have the opportunity to consent to have their data linked with provincial single-payer administrative health data. This linkage will enable access to additional contextual data that can help further identify trajectories outside the context of CAMH. The proposed linkage will allow for: 1) the evaluation of real world outcomes and service use, and 2) a comparison to other people with SSDs who may be receiving similar services in the community or have a SSD but are not connected with services. All linked data will be de-identified and held and analyzed at ICES. ICES is a prescribed

entity under section 45 Ontario's Personal Health Information Privacy Act. Linkages will occur across the following ICES data holdings: the Registered Persons Database (RPDB) which is a central population registry that contains basic demographic data on all people insured by OHIP; the Ontario Mental Health Reporting System (OMHRS) which contains data on all inpatient hospitalizations to adult mental health beds; the Canadian Institute for Health Information Discharge Abstract Database (CIHI-DAD) containing data on all acute care hospitalization and mental health hospitalizations before 2005; the National Ambulatory Care Reporting System (NACRS) which contains data on emergency department visits; outpatient physician billings from OHIP; the Ontario Drug Benefit (ODB) claims which contains data on medications dispensed through publicly funded program for eligible people; the Ontario Laboratories Information System (OLIS) containing laboratory data across the province, the Ontario Registrar General (ORG) for information on deaths; and the Ontario Marginalization Index (ON-Marg) which is an area-level deprivation index based on census data. Propensity scores will be calculated based on demographic and clinical covariates to enable the comparison between individuals receiving CAMH services, receiving community services, and those that are eligible but not utilizing services.

*3.3.2.5. EMR-based NLP and ML for the characterization of longitudinal clinical course and outcomes.* EMR data can be used to track the evolution of symptoms and treatments [181] and evaluate the impact of interventions on health outcomes [182]. Recent advances in natural language processing (NLP) also make it possible to analyze unstructured clinical data (i.e., consultation, progress, admission, and discharge notes), which can be combined with predictive modeling identify deteriorating trajectories of mental and physical health, physical functional status, as well as family, social, and spiritual supports, from structured and unstructured EMRs [183].

To retrospectively characterize the longitudinal clinical course and outcomes for study participants, our EMR analysis will be carried out in two stages. In the first stage, we will develop and evaluate an NLP methodology for analyzing clinical notes pertaining to patients with SSDs. To identify relevant topics or themes within the clinical notes, we will train topic models with Latent Dirichlet Allocation (LDA) [184]. LDA is an unsupervised ML algorithm which identifies the presence of topics in textual data based on the co-occurrences of words or phrases. The quality of emerging topics relies on the strategies used to preprocess text. However, research suggests that no one configuration of preprocessing rules is optimal across datasets and model types [185–187]. Therefore, we will explore various approaches to preprocessing the clinical notes, focusing on different ways to select the terms used to train the topic models (e.g., based on term frequency, term frequency-inverse document frequency weights, and named entity recognition). We will also train topic models on different clinical note types and sections, based on evidence that predicting clinical outcomes from medical notes may be improved when more relevant note types or parts are used [188]. To evaluate these approaches, we will examine associations between emerging topics and the available structured data (i.e., as a form of external validation), as well as perform human-in-the loop evaluations of topic coherence [189]. Our baseline model will be a topic model trained with LDA on all notes and all available terms, assuming symmetric Dirichlet prior distributions and using 2500 iterations of Gibbs sampling to learn 60 topics from each document. To examine how topic number ($k$) impacts the quality of modelling, we will evaluate a range of 25–75 topics. In all topic models, alpha (i.e., the Dirichlet prior over topic proportions per document) will be set to $50/k$, and beta (i.e., the symmetric prior over word distributions per topic) will be set to 200/the length of the vocabulary ($N$).

In the second stage, we will extract structured and unstructured EMRs of study participants. Structured data elements that overlap with measures utilized for prospective cohort characterization (described above) will include socio-demographic data, administrative data, clinical

data, and measures of functioning. To characterize clinical course and outcomes for study participants, we will train topic models on any available clinical records for these participants, using the best performing text preprocessing and modeling approaches from our evaluations of retrospective patient data in the first stage. Further evaluations of topic models at this stage may involve qualitative comparisons of emerging topics with clinical notes of study participants. Additionally, we will explore the value of deriving embeddings of clinical notes with large language models (e.g., ClinicalBERT; [190]), for predicting trajectory subtype membership (as described in Section 3.4). We will evaluate performance based on different layers (with the second to last layer as our baseline), as well as different aggregation methods (with concatenation used as a baseline method when aggregating from layers to tokens, and mean as a baseline method when aggregating from tokens to texts).

### 3.4. Data analysis

Descriptive statistics including point estimates and confidence intervals will be generated to summarize the data on all participants to understand the uni- and multi-dimensional characteristics of their distribution. Clustering of observations at multiple levels will be evaluated to provide guidance for selecting bias correction methods and multilevel modeling approaches. Full information maximum likelihood estimation [191] and multiple imputation methods [192] will be utilized when appropriate. As is typical in longitudinal research, we anticipate non-random missing data. Sensitivity analyses will be conducted to evaluate the impact of non-random missing data.

For Objective 1, latent growth mixture modeling [193] will be primarily applied to permit a person-centered approach for identification of homogeneous subgroups of individuals with distinct longitudinal outcome trajectories across our three co-primary outcome measures respectively. Fit indices (e.g. AIC and BIC), Lo-Mendell-Rubin likelihood ratio test, theoretical considerations, and clinical relevance will guide the decision of the optimum number of classes to represent the growth trajectories of personal recovery, disability, and community functioning. Given the expected heterogeneity of our sample particularly on age range and illness stages, it is important to assess measurement invariance and utilize methods (e.g, moderated nonlinear factor models) [194] that accommodate time-varying constructs over subsamples.

Further, we will adopt two multivariate longitudinal approaches, cross-lagged panel [195] and parallel processes [196] models, to model the longitudinal outcome measures simultaneously. The former is a more theory-based approach that can test directional effects that one variable has on another at different points in time; for example, earlier measures of functioning may predict later measures of disability. The latter could be used to associate multiple longitudinal trajectories with respect to their growth characteristics; for example, the slopes of growth trajectories of different outcomes may be associated with each other. In conjunction with growth mixture modeling under the accelerated longitudinal design, these methods could provide a more comprehensive view of the outcome measures of interest.

For Objective 2, we will associate the membership of the subgroups and thus the characteristics of these trajectories with sociodemographic and treatment factors, psychopathology, physical health, and biological measures at baseline by treating probabilistic membership as the dependent variable in the predictive models. Using structural equation modeling, we will combine the latent growth mixture model that determines the membership of the subgroups, measurement model that reduces the dimension of the predictors, and predictive model that links them into one step using simultaneous equations.

For our exploratory objective, we will investigate associations between temporal patterns emerging from topic modeling of unstructured EMRs, as well as embedding derived with large

language models with relevant measures collected prospectively from study participants. We will also use general linear model analyses as appropriate to investigate differential characteristics between longitudinal outcome trajectory subtypes. These temporal patterns and embeddings will also be examined as potential predictors for trajectory subtype membership in line with the methods employed for Objective 2.

We will conduct additional analyses to investigate the effect of sex and gender based on data captured through the PhenX toolkit demographics measure. For Objective 1, we will determine the longitudinal functional trajectories within sex and gender groups across three domains and examine if there are substantial differences between sex and gender groups and if necessary, test Hypotheses 1a and 1b within sex and gender groups as a supplement analysis. For Objective 2, in addition to examining sex and gender as a main effect in predicting functional trajectories at the person level, we will further assess if it interacts with other sociodemographic, treatment factors, psychopathology, physical health, and biological measures in the predictive models.

**3.4.1. MRI data analysis.** *3.4.1.1. Structural and resting state functional MRI analysis.* These scans will be preprocessed together using the CAMH-developed and publicly released Ciftify pipeline [197]. This pipeline incorporates workflows from FreeSurfer [198] for structural preprocessing and analysis and fMRIPrep for functional preprocessing [199].

*3.4.1.2. Diffusion MRI analysis.* After preprocessing (including eddy current correction, nonlinear EPI distortion correction filtering, and tensor estimation), we will calculate diffusion measures including fractional anisotropy, axial diffusivity, radial diffusivity, and mean diffusivity to investigate white matter integrity and connectivity. Researchers at CAMH have recently shown multi-shell diffusion combined with novel non-tensor models can reliably generate measures of free water, diffusion kurtosis, the orientation dispersion index (ODI), and neuritic density index (NDI) in gray matter [200]. Moreover, ODI has been associated with cognitive performance in aging and changes in NDI in psychiatric disorders [201]. These measures will be combined with measures from the structural pipeline to drive multi-modal profiles of "brain-age" trajectories.

*3.4.1.3. Extracting fMRI features with high reliability.* All BOLD fMRI will be concatenated together to increase the reliability of the BOLD connectivity signals [202]. Under this framework, we anticipate that we may see r > 0.4 for > 50% of individual edges [202].

*3.4.1.4. Abdominal imaging.* In house scripts will be used to obtain liver fat percentage. Selective thresholding along with automated and supervised contouring will be utilized to estimate subcutaneous (SAT) and visceral fat (VAT). SAT and VAT for each participant will be measured by segmenting the appropriate fat pixels on each of the acquired slices and then measuring the total volume of the segmented pixels in mL.

**3.4.2. Statistical power and sample size.** We have assessed statistical power for the following targeted analyses aimed to a) identify distinct longitudinal functional trajectories of individuals with SSDs; b) establish predictive models to associate personal factors with growth trajectories; and c) provide feasibility indices for future interventions. Assuming that we will have approximately four distinct longitudinal trajectories, through a Monte-Carlo study, we concluded that we have sufficient power (0.80) to correctly enumerate the classes when they are evenly distributed and reasonable power (0.68) when they are not. In the latter, we assumed that the smallest class only takes up 15% of the total sample. Further, we concluded that we have abundant power (0.91) to detect linear trends of small change over time (effect size of 0.20 between two consecutive waves) even for the smallest class (n = 112). To identify associations between predictors (e.g., social determinants) and group membership of the different trajectories, we also anticipate to have sufficient power (0.82 to 0.85) for detecting a small odds ratio (OR = 1.68) depending on the distribution of the predictors. In addition, this

study will provide reliable estimates for feasibility indices for future interventions. For all aforementioned power estimation, we assumed 25% overall attrition and used .05 as the significance level.

## 4. Data management protocols

Data will be collected on REDCap forms (electronic database) and housed at CAMH. Labkey will be used to support tracking and management of biological samples and genetic analysis data. All data will be linked via each participant's unique study ID. Data collection will take place either in-person or virtually through the use of the Cisco Webex videoconferencing platform. All MRI data will be uploaded to XNAT and automated pipelines used for standardized processing and QA/QC procedures. A protocol data collection schedule will be used to monitor participant progress, missing assessments, and other protocol deviations during the study. Data entry screens will incorporate range checks or lists of valid responses for each item to ensure accurate data entry. Forms with missing or invalid data in key identifying fields will be referred back to raters for correction before entry. Other missing or invalid data will not prevent the form from being entered, but will be flagged for correction. Queries will be built around data entry and reports to review adverse events. Participant confidentiality will be maintained by restricting study data access to specified study personnel. At the conclusion of the study, all study data will be archived and retained for the full period required by regulations and according to CAMH SOPs.

For retrospective patient data obtained from health records, all structured and unstructured EMRs will be extracted, processed, and analyzed in accordance with established protocols and safeguards. These protocols were co-developed with CAMH's Research Ethics Board and Privacy Department to protect patient privacy. In brief, they involve hospital data warehouse staff de-identifying unstructured EMRs and extracting numerical features from the text data before it is shared with the study team. Because clinical records can contain highly contextualized personal and identifiable information, these protocols ensure that the sensitive data (in its raw form) never leaves dedicated hospital clinical servers and is never accessed directly by the study team. EMRs in the second stage will be accessed and analyzed with consent from study participants.

### 4.1. Open science and data sharing

The PREDICTS Study will implement principles of open science by utilizing processes and data governance being established by the CAMH BrainHealth Databank to enable and promote data sharing and reuse with researchers at CAMH and around the world. This includes implementation of study data management processes to ensure that study data adhere to the FAIR data principles–Findable, Accessible, Interoperable, and Reusable, as well as the collection of common data elements to facilitate data standardization and harmonization as appropriate. To support data sharing, a PREDICTS cohort explorer dashboard will be created to allow users to explore available data and formulate data access requests. All information entered into these databases will be free of identifying information. Data access to researchers conducting secondary studies will be granted once appropriate approval has been obtained, as defined in the BrainHealth Databank Data Access Policy.

## 5. Summary

Despite advances in understanding and treating SSDs, enduring disability continues to be the hallmark of these serious and persistent mental illnesses. The proposed work aims to advance the early identification and prediction of individual-level functional and recovery trajectories

for people experiencing SSDs and explore the core contributors to these trajectories. Findings from this study may set the stage for the development of future interventions targeted to one's illness trajectory to both promote and improve functional outcomes, recovery, and community functioning in different settings. From a neuroscience and biological perspective, this work may be able to identify biological and metabolic profiles, physical health parameters, and structural and functional properties of brain circuits that are associated with, and which may predict different illness and functional trajectories.

This study also has the potential to impact the administration of clinical care. Based on data from this study, measurement-based approaches and future family and patient engagement will provide templates for new co-design care models to further improve best practice.

Furthermore, this initiative will help address the huge gaps in physical care that drive morbidity and premature mortality in this population. We anticipate that these efforts will open up the potential for system level change to ensure individuals are treated from all health-related (i.e. physical and mental) perspectives, also bridging medical and mental health care for individuals with SSDs.

## Supporting information

**S1 File.**
(DOCX)

## Author Contributions

**Conceptualization:** Sri Mahavir Agarwal, Margaret Hahn, George Foussias.

**Funding acquisition:** Margaret Hahn, George Foussias.

**Investigation:** Sri Mahavir Agarwal, Joel Dissanayake, Ofer Agid, Christopher Bowie, Noah Brierley, Araba Chintoh, Vincenzo De Luca, Andreea Diaconescu, Philip Gerretsen, Ariel Graff-Guerrero, Colin Hawco, Yarissa Herman, Sean Hill, Kathryn Hum, Muhammad Omair Husain, James L. Kennedy, Michael Kiang, Sean Kidd, Nicole Kozloff, Marta Maslej, Daniel J. Mueller, Farooq Naeem, Nicholas Neufeld, Gary Remington, Martin Rotenberg, Peter Selby, Ishraq Siddiqui, Kate Szacun-Shimizu, Arun K. Tiwari, Shanthos Thirunavukkarasu, Wei Wang, Joanna Yu, Clement C. Zai, Robert Zipursky, Margaret Hahn, George Foussias.

**Methodology:** Sri Mahavir Agarwal, Margaret Hahn, George Foussias.

**Project administration:** Margaret Hahn, George Foussias.

**Supervision:** Margaret Hahn, George Foussias.

**Writing – original draft:** Sri Mahavir Agarwal, Joel Dissanayake, Margaret Hahn, George Foussias.

**Writing – review & editing:** Sri Mahavir Agarwal, Joel Dissanayake, Ofer Agid, Christopher Bowie, Noah Brierley, Araba Chintoh, Vincenzo De Luca, Andreea Diaconescu, Philip Gerretsen, Ariel Graff-Guerrero, Colin Hawco, Yarissa Herman, Sean Hill, Kathryn Hum, Muhammad Omair Husain, James L. Kennedy, Michael Kiang, Sean Kidd, Nicole Kozloff, Marta Maslej, Daniel J. Mueller, Farooq Naeem, Nicholas Neufeld, Gary Remington, Martin Rotenberg, Peter Selby, Ishraq Siddiqui, Kate Szacun-Shimizu, Arun K. Tiwari, Shanthos Thirunavukkarasu, Wei Wang, Joanna Yu, Clement C. Zai, Robert Zipursky, Margaret Hahn, George Foussias.

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
