## [Editor Report · Decision Letter 0]

16 Mar 2023

PONE-D-23-05144Characterization and Prediction of Individual Functional Outcome Trajectories in Schizophrenia Spectrum Disorders (PREDICTS Study): Study ProtocolPLOS ONE

Dear Dr. Agarwal,

Thank you for submitting your manuscript to PLOS ONE. After careful consideration, we feel that it has merit but does not fully meet PLOS ONE’s publication criteria as it currently stands. Therefore, we invite you to submit a revised version of the manuscript that addresses the points raised during the review process. Thank you for the opportunity to review your study protocol.

This is an intriguing and timely proposal – and I do think it has its merits and should be considered. Nevertheless, I have some concerns, they are brief, to be solved before proceed with your study.

Please details your Methods: sample size calculation, power analysis, eligibility criteria (re. medication use, duration of the disease, and other aspects in a separate section for eligibility and exclusion etc. criteria), what are the scales for psychopathology, all of the Cronbach values for scales, details of the analyses (effect sizes, CIs, whether subgroups analysis would be conducted and why), details of the machine learning (which algorithm, how many leaves, the depth), details of the other techniques (refreshing rate, the slices, number of the slices and how they will be re-constructed);Also please detail, in plain text, the arms of the protocol and how to handle with missing data, the expected effect to yield, and the relevant specifications;Although not directly related, for now, I’d strongly recommend to use SPIRIT for building sections;Finally, please consider text overlapping or the possibility of text duplication – at least the system is reporting it. If you had this data on OSF or something, please place in text. If the text or parts of It (> 20% overlapping) then place in some section of the text.==============================

We look forward to receiving your revised manuscript.

Kind regards,

Thiago Fernandes, PhD

Academic Editor

PLOS ONE

Journal Requirements:

2. Our internal editors have looked over your manuscript and determined that it is within the scope of our Reproducibility and Replicability in Neuroscience and Mental Health Research Call for Papers. The Collection will encompass a diverse and interdisciplinary set of protocols and research articles adhering to transparent and reproducible reporting practices in the areas of clinical psychology, psychiatry, mental health, and neuroscience. Additional information can be found on our announcement page: https://collections.plos.org/call-for-papers/reproducibility-and-replicability-in-neuroscience-and-mental-health-research/. If you would like your manuscript to be considered for this collection, please let us know in your cover letter and we will ensure that your paper is treated as if you were responding to this call. If you would prefer to remove your manuscript from collection consideration, please specify this in the cover letter.
---

## [Author Response · Author response to Decision Letter 0]

1 Jun 2023

Response to Reviewer Comments

Reviewer: 1

Comments to the Author

1. Please details your Methods: sample size calculation, power analysis, eligibility criteria (re. medication use, duration of the disease, and other aspects in a separate section for eligibility and exclusion etc. criteria), what are the scales for psychopathology, all of the Cronbach values for scales, details of the analyses (effect sizes, CIs, whether subgroups analysis would be conducted and why), details of the machine learning (which algorithm, how many leaves, the depth), details of the other techniques (refreshing rate, the slices, number of the slices and how they will be re-constructed);

- Thank you for your comments. With regards to the methodology section for this study protocol, specifically eligibility criteria, we have highlighted both the inclusion and exclusion criteria and made it more explicit as it may have not been before. Individuals will be included in this study if they are: 1) 16 years of age and older; 2) have a DSM-5 diagnosis of and SSD or other disorder with psychotic features such as BPD, MDD, and/or substance abuse/medication-induced psychotic disorder; and 3) have adequate fluency in English to participate in clinical care without the need of a translator. The edits can be seen on page 11. Regarding the scales for psychopathology, a table is listed for all of the scales that we will use throughout the study along with additional information regarding internal validity (Cronbach alpha values) beginning on page 13. For further clarification on machine learning, we are using ML in our study to analyze the clinical notes (i.e., topic modelling with Latent Dirichlet Allocation, an unsupervised ML algorithm, and deriving embeddings with large language models), and not to train ML algorithms on historical/training data to predict outcomes in prospective/test data. To avoid confusion, we now refer to this approach as NLP based (and not NLP and ML based). We have also added more detail related to our NLP methods (i.e., preprocessing, LDA parameters, large language modelling parameters) in hopes that this clarifies our approach. Edits to this section can be seen on pages 16 and 17. Edits to other techniques (MRI parameters) regarding the methodology section can be seen on page 15. 

2. Also please detail, in plain text, the arms of the protocol and how to handle with missing data, the expected effect to yield, and the relevant specifications;

- Thank you for this comment. We have edited the information pertaining to the expected data and the analysis on pages 17-19 to explicitly outline relevant specifications. 

3. Although not directly related, for now, I’d strongly recommend to use SPIRIT for building sections;

- Thank you for this suggestion. We have adapted our paper to fit the SPIRIT guidelines for building sections. In addition, we have changed the format of our citations/references to meet the journal guidelines. 

4. Finally, please consider text overlapping or the possibility of text duplication – at least the system is reporting it. If you had this data on OSF or something, please place in text. If the text or parts of It (> 20% overlapping) then place in some section of the text.

- Thank you for this suggestion. We have put this manuscript through an educational software to ensure there is limited text overlapping. Through the software we have identified that references are what are causing the overlap of >20%. Once references were removed from the document, the overlap score was below 20%.

---

## [Editor Report · Decision Letter 1]

26 Jun 2023

Characterization and Prediction of Individual Functional Outcome Trajectories in Schizophrenia Spectrum Disorders (PREDICTS Study): Study Protocol

PONE-D-23-05144R1

Dear Dr. Foussias,

We’re pleased to inform you that your manuscript has been judged scientifically suitable for publication and will be formally accepted for publication once it meets all outstanding technical requirements.

Kind regards,

Thiago P. Fernandes, PhD

Academic Editor

PLOS ONE
---

## [Editor Report · Acceptance letter]

11 Sep 2023

PONE-D-23-05144R1 

Characterization and Prediction of Individual Functional Outcome Trajectories in Schizophrenia Spectrum Disorders (PREDICTS Study): Study Protocol 

Dear Dr. Foussias:

I'm pleased to inform you that your manuscript has been deemed suitable for publication in PLOS ONE. Congratulations! Your manuscript is now with our production department. 

Kind regards, 

on behalf of

Dr. Thiago P. Fernandes 

Academic Editor

PLOS ONE